# Well-conditioned Spectral Transforms for
# Dynamic Graph Representation

**Bingxin Zhou**[*]
The University of Sydney, Shanghai Jiao Tong University
bzho3923@uni.sydney.edu.au

**Xinliang Liu**[*]
King Abdullah University of Science and Technology
xinliang.liu@kaust.edu.sa

**Yuehua Liu**
Shanghai Jiao Tong University
liuyh1214@gmail.com

**Yunying Huang**
The University of Sydney
yunying.huang@sydney.edu.au

**Pietro Liò**
University of Cambridge
pl219@cam.ac.uk

**Yu Guang Wang**
Shanghai Jiao Tong University
yuguang.wang@sjtu.edu.cn

## Abstract

This work establishes a fully-spectral framework to capture informative long-range temporal interactions in a dynamic system. We connect the spectral transform to the low-rank self-attention mechanisms and investigate its energy-balancing effect and computational efficiency. Based on the observations, we leverage the adaptive power method SVD and global graph framelet convolution to encode time-dependent features and graph structure for continuous-time dynamic graph representation learning. The former serves as an efficient high-order linear self-attention with determined propagation rules, and the latter establishes scalable and transferable geometric characterization for property prediction. Empirically, the proposed model learns well-conditioned hidden representations on a variety of online learning tasks, and it achieves top performance with a reduced number of learnable parameters and faster propagation speed.

## 1 Introduction

Dynamic graphs appear in many scenarios, such as pandemic spread [1, 2], social media [3, 4], physics simulations [5, 6], and computational biology [7, 8]. Learning dynamic graph properties, however, is a challenging task when both node attributes and graph structures evolve over time.

Many existing dynamic graph representation learning methods start from embedding the sequence of non-Euclidean graph topology to feed into recurrent networks [9–12]. Such a straightforward design assumes a discrete nature of input graphs. Graph snapshots are sliced at a sequence of fixed time steps, leaving the evolution of events on nodes and/or edges unobserved. Later, the memory module [4, 13] establishes a natural generalization of the learning procedure to continuous-time dynamic graphs (CTDGs), which encodes previous states for an event to its latest states. Consequently, a graph slice describes the past dynamics with implicitly encoded long-short term memory on node attributes.

Nevertheless, the memory module, e.g., recurrent neural networks [14] or gated recurrent unit [15], has trouble tracking the full picture of graph evolving, as it reserves long-term interactions in a most implicit way. Accessing the encoded message inside the black box becomes extremely hard. Alternatively, TRANSFORMER [16] enhances the long-range memory for sequential data, and it has received tremendous success in language understanding [17, 18] and image processing [19, 20]. In

---

[*]Equal contribution.

B.Zhou et al., Well-conditioned Spectral Transforms for Dynamic Graph Representation. *Proceedings of the First Learning on Graphs Conference (LoG 2022)*, PMLR 198, Virtual Event, December 9–12, 2022.

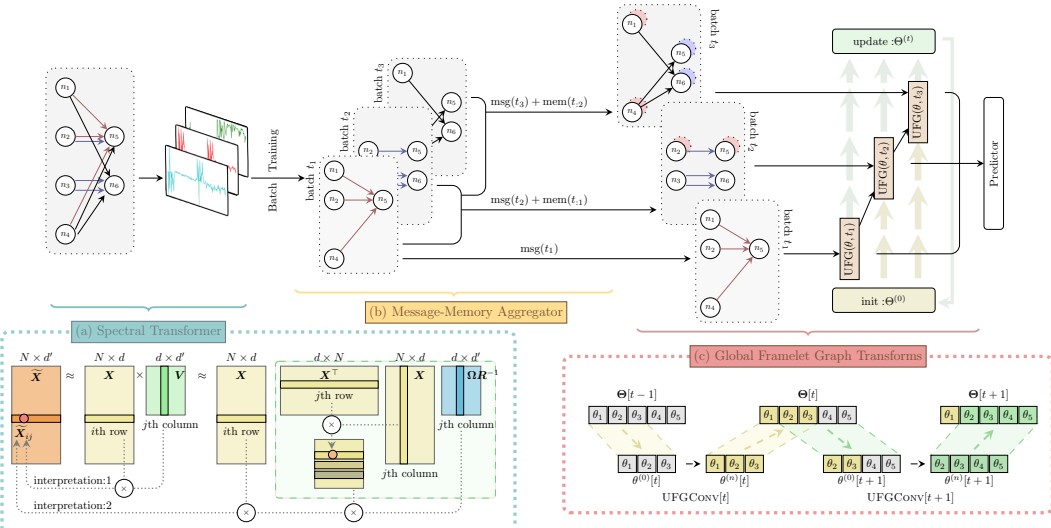

**Figure 1:** Illustrative SPEDGNN for learning continuous-time dynamic graphs (CTDGs). (a) A *spectral transform* with adaptive power method SVD processes the long-range time-dependency of the input to the spectral domain. (b) The continuous embedding is then divided in a *message-memory* module with enhanced short-term interactions. (c) Finally, a *global framelet graph convolution* with multi-scale operators forms well-conditioned graph representations for prediction tasks.

particular, the self-attention mechanism learns pair-wise event similarity scores in the entire range of interest. It retrieves a contextual matrix of full-landscape relationships to preserve the long-term dependency of tokens (or events). However, at the cost of comprehensiveness, the rapid growth of the sequence length can easily escalate the complexity of computation and memory. While the attention operation can be efficiently approximated by some low-rank representation [21–24], it loses the expressivity at the same time.

This work provides a fully spectral-based solution for learning the representations of long-range CTDGs. First, an efficient *spectral transform* enhances the memory encoding of continuous events by extracting pairwise nonlinear relationships in time and feature dimensions. A *global spectral graph convolution* with fast framelet transforms [25] then characterizes node-wise interactions in a sequence of graphs. The proposed design tackles the two identified problems in learning CTDGs. In particular, we show that the power method singular value decomposition (SVD) is an efficient and effective implementation of the low-rank self-attention scheme. It not only fast captures the long-term evolving flow of the input events, but also preserves more even energy in the extracted pivotal components of the temporal observations. Such a design prevents ill-conditioned graph hidden representations, which results in an easier-to-fit smooth decision boundary for network training. In the final layer, the undecimated framelet-based spectral graph transform in graph representation learning commits sufficient scalability via its multi-level representation of the structured data.

We investigate the relationship of spectral transforms and feed-forward propagation, and design ***Spectral Dynamic Graph Neural Network*** (SPEDGNN) for efficient and effective dynamic graph representation. The design network architecture captures temporal features and graph structure in CTDGs in the spectral domain. Through efficient spectral self-attention and multi-scale graph convolution, expressive hidden representations of batch events are embedded in linear complexity (proportional to the number of events). The well-conditioned final embeddings are separable by a smooth decision boundary with less main information loss.

## 2 Spectral Transform for Long-range Sequence

This section introduces the notion of spectral transform and discusses how it fixes the ill-conditioning problem and its connection to the self-attention mechanism.

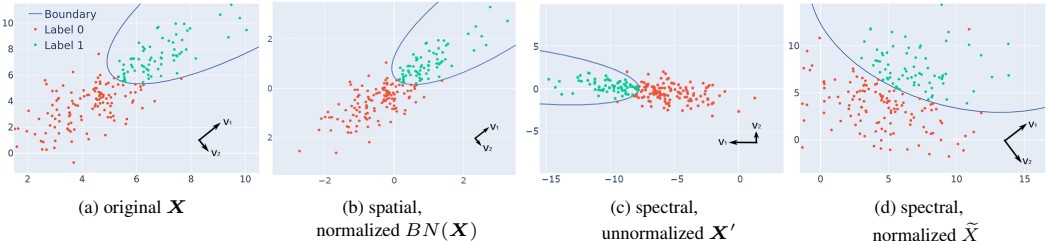

(a) original $\boldsymbol{X}$     (b) spatial, normalized $BN(\boldsymbol{X})$     (c) spectral, unnormalized $\boldsymbol{X}'$     (d) spectral, normalized $\widetilde{X}$

**Figure 2:** A toy example of binary classification. The 2-dimensional data are sampled from $\mathcal{N}(\mu, \sigma)$. The direction and length of $\{v_1, v_2\}$ illustrate two eigenvectors and eigenvalues of the feature. Artificial labels are created by a decision boundary with large curvature. Both (b) batch normalized spatial representations and (c) unnormalized spectral coefficients fail to flatten the boundary. In contrast, (d) normalized spectral representations have a closer-to-1 condition number, creating a smooth decision boundary that is easier for a classifier to fit.

**Definition 1.** A *spectral transform* projects sample observations $\boldsymbol{X} \in \mathbb{R}^{N \times d}$ from an unknown function space to a spectral domain with a (set of) orthonormal basis $\boldsymbol{\Phi}$: $\boldsymbol{X}' := \boldsymbol{X}\boldsymbol{\Phi}$. The new representation $\boldsymbol{X}'$ summarizes the prior knowledge of observations $\boldsymbol{X}$ for perfect reconstruction.

### 2.1 Choices of the Spectral Basis

The properties of a formulated spectral transform are determined by $\boldsymbol{\Phi}$. For instance, the singular vector matrix of $\boldsymbol{X}$ from QR decomposition or singular value decomposition (SVD) extracts principal components of the function space. Fourier transforms process time-domain signals to the frequency domain to distill local-global oscillations. Such a transform summarizes the observations in a new coordinate system to reflect desired properties of a sequential $\boldsymbol{X}$, such as sparsity or noise separation.

To better understand how spectral transform benefits inferring the true function space, consider an example unitary transform by orthonormal bases of SVD. Denote the raw signal input as a matrix $\boldsymbol{X} \in \mathbb{R}^{N \times d}$. It can be factorized by $\boldsymbol{X} = \boldsymbol{U}\boldsymbol{\Sigma}\boldsymbol{V}^\top$ with two orthonormal bases $\boldsymbol{U} \in \mathbb{R}^{N \times N}$ and $\boldsymbol{V} \in \mathbb{R}^{d \times d}$. The two orthonormal bases span the row and column spaces of $\boldsymbol{X}$, and they both can project $\boldsymbol{X}$ to a spectral domain. For instance, the spectral coefficients $\boldsymbol{X}' = \boldsymbol{X}\boldsymbol{V}$ are the projection of aggregated features under the basis $\boldsymbol{V}$.

### 2.2 Transforming towards Balanced Energy

A key motivation to perform the spectral transform on a time-dependent long-range sequence is to amend the highly-imbalanced energy distribution of the original feature space. We pay special attention to the cases when the expressivity of latent representations is hurt, i.e., the detailed information with small energy in the original feature space is smoothed out. In practice, such small-energy details can be pivotal to distinguishing different entities, and ignoring them not only removes local noise but also eliminates potentially useful messages. For instance, tumor cells generally live within a small area and it has considerably small energy in a medical image. Smoothing these features could result in problems in pathology diagnosis.

Amending the energy distribution, however, can be tricky to conduct in the features' original domain. Figure 2 demonstrates a two-dimensional toy example. The sample distribution in Figure 2(a) concentrates the most variance in a certain direction with eigenvalues of sample variance $\sigma = \{9, 2\}$. An instant normalization in the same domain, such as BatchNorm [26] in Figure 2(b), reshapes the sample distribution. However, the energy is still centralized in $\boldsymbol{v}_1$'s direction. As a result, the two classes (colored in red and green) can only be divided by a decision boundary that has a large curvature. It is difficult to fit by classifiers such as an MLP-based model, which tends to fit smooth flat curves. Meanwhile, Figure 2(c) illustrates the unnormalized spectral representation of the original data $\boldsymbol{X}' := \boldsymbol{X}\boldsymbol{V} = \boldsymbol{U}\boldsymbol{\Sigma}$, which projects $\boldsymbol{X}$ to a new coordinate set by the transformation $\boldsymbol{V}$. As shown in Figure 2(d), normalizing the new coordinates in the same spectral domain by $\widetilde{\boldsymbol{X}} := \boldsymbol{X}'\mathrm{diag}(c_1, c_2)\boldsymbol{V}^T$ results in an easy-to-fit flat decision boundary.

The spectral transform allows balancing features' energy and truncating local noise, if necessary, simultaneously. To circumvent singular decomposition, we consider an efficient approximation

that relies on matrix products of $X$. We can regard $U$ (from $X = U\Sigma V^T$) to be close to the orthonormal basis $Q$ (from QR decomposition $X = QR$). Power method SVD [27] suggests a better approximation to $U$, which is the orthonormal basis $\tilde{Q}$ from $X(X^\top X)^q = \tilde{Q}\tilde{R}$, i.e.,

$$U \approx \tilde{Q} = X(X^\top X)^q \tilde{R}^{-1}. \tag{1}$$

The normalized features $\widetilde{X}$ is therefore approximated by including a proper diagonal matrix $C$ in (1), i.e., $\widetilde{X} \approx X(X^\top X)^q \tilde{R}^{-1} C$. However, it is computationally expensive to directly invite the orthogonal factor $\tilde{R}$ to participate in every propagation of the neural network, as every QR decomposition involves a Gram-Schmidt algorithm. To balance the computational cost and the expressivity, we transfer the progressive update on $\tilde{R}$ to $C$. We let

$$\widetilde{X} \approx X(X^\top X)^q W, \tag{2}$$

where $W = \tilde{R}^{-1} C$ constitutes a fixed $\tilde{R}$ and a learnable diagonal $C$ initialized as an identity matrix. Consequently, $\widetilde{X}$ supports matrix computations and network propagation.

Compared to the conventional truncated SVD that ranks the orthonormal basis by singular values, the learnable spectral transform in (2) conducts a data-driven principal component distillation and normalization simultaneously. The learnable projection $W$ plays a similar role to the computationally intensive Gram-Schmidt orthonormalization that summarizes the entity features to a set of spectral coefficients and ranks them adaptively by their importance to the underlying application. Such a learning scheme prevents important rare patterns from being removed due to their small energy.

### 2.3 Connecting Adaptive Spectral Transform to Self-Attention

Aside from preserving small-energy rare patterns as other spectral transforms, the adaptive SVD-based spectral transform is also closely connected to linear self-attention mechanisms [21, 23, 28]. For $X \in \mathbb{R}^{N \times d}$, a self-attention layer reads

$$X_{\text{attn}} := (Q_a K_a^\top V_a)/\sqrt{d_k}, \tag{3}$$
$$\text{where } Q_a := XW_Q, \ K_a := XW_K, \ V_a := XW_V.$$

The three square matrices $Q_a$ (query), $K_a$ (key) and $V_a$ (value) learn basis functions at an identical size of $N \times d$. The learning cost drops significantly when $N \gg d$, as a smaller number of parameters are required to approximate. In comparison, conventional self-attentions activate the context mapping matrix $\text{softmax}(Q_a K_a^\top / \sqrt{d_k}) \in \mathbb{R}^{N \times N}$. The calculation order of (3) is thus required strictly from left to right, which rejects efficient matrix computations due to the inevitable $N$-dimension.

To understand the intrinsic connection between linear self-attention in (3) and power method SVD, rewrite $X_{\text{attn}}$ as a function of $X$, i.e.,

$$X_{\text{attn}} = XW_Q W_K^\top X^\top XW_V = XW_1 X^\top XW_2 \tag{4}$$

with $W_Q W_K^\top = W_1$ and $W_V = W_2$. Compared to (2), a linear self-attention step in (4) is a special implementation that approximates a 1-iteration QR approximation of the SVD basis. To approach the power of $q$ iterations adaptive power method SVD, a number of $q$-layer linear self-attention is required. Moreover, both (2) and (4) aggregate row-wise variation and summarizes a low-rank covariance matrix of $X$ with $X^\top X$. However, (2) provides an efficient concentration to large-mode tokens while truncating out noises. For an extremely long sequence of input $X_N \in \mathbb{R}^{N \times d}(N \gg d)$, $X_N^\top X_N \in \mathbb{R}^{d \times d}$ in (2) completes the main calculation at a significantly small cost. This cost-efficient technique is important for scalable learning tasks such as time-series data learning, where the length of an input sequence could explode easily.

## 3 Spectral Transforms for Dynamic Graphs

In this section, we expand the long-range sequence of interest to an additional dimension of topology and practice the spectral transform on dynamic graphs. We validate the efficiency and effectiveness of the spectral transform framework by dynamic graph representation learning.

## 3.1 Problem Formulation

A static undirected graph is denoted by $\mathcal{G}_p = (\mathbb{V}_p, \mathbb{E}_p, \boldsymbol{X}_p)$ with $n = |\mathbb{V}_p|$ nodes, where its edge connection is described by an adjacency matrix $\boldsymbol{A}_p \in \mathbb{R}^{n \times n}$ and the node features are stored in $\boldsymbol{X}_p \in \mathbb{R}^{n \times d}$. A graph convolution finds a hidden representation $\boldsymbol{H}_p$ of the structure $\boldsymbol{A}_p$ and the node feature $\boldsymbol{X}_p$. When $\boldsymbol{A}_p$ and/or $\boldsymbol{X}_p$ change with time, $\mathcal{G}_p$ is called a *dynamic graph*. Dynamic graph representation learning finds the hidden representation $\boldsymbol{H}_p$ of each $\mathcal{G}_p$ from a sequence of graphs $\mathbb{G} = \{\mathcal{G}_p\}_{p=1}^{P}$ where each $\mathcal{G}_p = (\mathbb{V}_p, \mathbb{E}_p, \boldsymbol{X}_p)$. Depending on the particular prediction task, $\boldsymbol{H}_p$ can be processed for label assignments. For example, link prediction forecasts the pair-wise connection of nodes in a graph, and node classification completes unlabeled nodes.

*Continuous-time dynamic graphs* (CTDG) is a general and complicated genre of dynamic graphs. An arbitrary observation of a CTDG is recorded as a tuple of *(event, event type, timestamp)*. The *event* recorded at a specific *timestamp* is described by a feature vector, and the *event type* can be one of edge addition/deletion, or node addition/deletion. Training an adequate model for CTDG, however, can be challenging. Compared to a static graph, the complete architecture of a CTDG is revealed sequentially during training. A powerful design for graph embedding is thus required to interpret the connection between the next graph with historical graph snapshots. In comparison to discrete-time dynamic graphs, the consecutive activity recording behavior allows CTDGs to capture the event flow of the entire graph so that the information loss is minimized.

To this end, we propose to employ *adaptive temporal spectral transform*s to encode the long-range evolution of the graph dynamics to normalized spectral coefficients $\widetilde{\boldsymbol{X}}$ for a minimum loss of energy. The short-term interaction is enhanced by employing a *message-memory* module [4, 13] then partitioned evenly into a sequence of subgraphs of interactive nodes, where the node attributes encode its present and recent status. Next, the graph topology is embedded by another spectral-based graph network, i.e., a *global spectral graph convolution*, to find a well-conditioned hidden representation for the final prediction task. We now explain the two spectral-based transforms in detail.

## 3.2 Adaptive Temporal Spectral Transform

The long-range time-dependency is encoded with *adaptive power method SVD* as a particular implementation of the temporal spectral transform. As briefed in Section 2, it takes a similar role as the traditional self-attention in feature extraction but is equipped with additional scalability and reliability. We focus on the transform and ignore the adaptive normalization for conciseness.

For an event sequence $\boldsymbol{X} \in \mathbb{R}^{N \times d}$, we look for its expressive low-dimensional projection $\boldsymbol{X}'$ in the spectral domain that i) summarizes the principal patterns of $\boldsymbol{X}$, and ii) is immune to minor disturbances. Analogous to self-attentions, the spectral encoder assigns a matrix of similarity scores to $\boldsymbol{X}$, and it follows an explicit update rule to establish a traceable learning process. The main patterns from both event attributes and time dimensions are summarized in spectral coefficients $\boldsymbol{X}'$ (cyan box in Figure 1). Below we explain the two interpretations of such transforms.

**Interpret 1. Spectral coefficients $\boldsymbol{X}' \approx \boldsymbol{X}\boldsymbol{V}$ extract information in feature dimension.** By definition $\boldsymbol{X} := \boldsymbol{U}\boldsymbol{\Sigma}\boldsymbol{V}^{\top}$, SVD stores the factorized features (in columns of $\boldsymbol{V}$) and temporal shifts (in rows of $\boldsymbol{U}$). For low-rank or noisy input, *truncated SVD* [29] extracts stable main patterns by $\boldsymbol{X}' \approx \boldsymbol{X}\boldsymbol{V} \in \mathbb{R}^{N \times d'} (d > d')$. Specifically, the transformed $\boldsymbol{X}'$ is projected by $\boldsymbol{V}$ to a new space of the most effective feature representation. For instance, the $j$th feature of the $i$th transformed event $\boldsymbol{X}'_{ij} = \boldsymbol{X}_{i,:}\boldsymbol{V}_{:,j}$ concretes $\boldsymbol{X}_i$ to a coefficient following the projection of the $j$th factorized feature.

**Interpret 2. Spectral coefficients $\boldsymbol{X}' \approx \boldsymbol{X}(\boldsymbol{X}^{\top}\boldsymbol{X})\boldsymbol{R}^{-1}$ aggregates information of time dimension.** We focus on the simplest case of iteration $q = 1$ for illustration purpose, i.e., $\boldsymbol{X}^{\top}\boldsymbol{X}\boldsymbol{R}^{-1}$ is a one-step approximation of $\boldsymbol{V}$. For instance, the $j$th element in the $j$th row of $\boldsymbol{X}^{\top}\boldsymbol{X}$ is from $\boldsymbol{X}_{:,j}$ that covers the whole time interval. The consequent covariance matrix $\boldsymbol{X}^{\top}\boldsymbol{X} \in \mathbb{R}^{d \times d}$ summarizes the column-wise linear relationship of $\boldsymbol{X}$, i.e., the change of attributes over time. Transforming $\boldsymbol{X}$ by this similarity matrix establishes a new presentation with the all-time temporal correlation of attributes. For $q > 1$, the approximation takes linear adjustments via $\boldsymbol{R}^{-1}$ and concentrates high energies on more expressive modes with the same fundamental format of the covariance matrix $\boldsymbol{X}^{\top}\boldsymbol{X}$.

### 3.3 Memory-Message Aggregator

Given an event $e_i[t]$ at time $t$ with respect to node $n_i$, we name it a *message* of $n_i$ at time $t$, denoted as $\mathbf{msg}(e_i[t])$. In addition, if the node was previously recorded active, we use $\mathbf{mem}(e_i[:t])$ to represent the past information, or *memory*, of $n_i$ prior to time $t$. The memory module $\mathbf{mem}(\cdot)$ refreshes constantly with the latest messages to capture the dynamic nature of graph interactions. When a new event $e_i[t]$ is recorded at time $t$, the memory updates to $\mathbf{mem}(e_i[t]) = f(\mathbf{mem}(e_i[:t]), \mathbf{msg}(e_i[t]))$ with a trainable function $f(\cdot)$. Depending on when the node $i$ was previously recorded, the last memory can be found before $t-1$. Also, it is possible to recall memories from multiple steps away.

We thus describe $n_i[t]$'s state by its hidden memory $h_i[t]$ at time $t$, which concatenates $\mathbf{msg}(e_i[t])$ and $\mathbf{mem}(e_i[:t])$, i.e., $h_i(t) = \mathbf{concat}(\mathbf{msg}(e_i[t])) \| \mathbf{mem}(e_i[:t])))$. The embedding for an interactive event $e_{ij}(t)$ between two nodes $n_i$ and $n_j$ is similar, which reads $h_i(t) = \mathbf{concat}(\mathbf{msg}(e_{ij}[t]) \| \mathbf{mem}(e_i[:t]) \| \mathbf{mem}(e_j[:t]))$.

### 3.4 Global Framelet Graph Transforms

We leverage global spectral graph convolutions to extract multi-level and multi-scale features in scalable graph representation learning. The vanilla framelet graph convolution (UFGCONV) [25] implements fast framelet decomposition and reconstruction for efficient static graph topology embedding (See Appendix B). Working in the framelet domain has been proven robust to local perturbations and circumvents over-smoothing with Dirichlet energy preservation [30, 31]. For CTDGs, we propose a global version of framelet transforms to perform multi-scale robust graph representation learning.

Formally, the *graph framelet convolution* defines in a similar manner to any typical spectral graph convolution layer that $\boldsymbol{\theta}_p \star \boldsymbol{X}_p = \boldsymbol{\mathcal{V}}_p \mathrm{diag}(\boldsymbol{\theta}_p) \boldsymbol{\mathcal{W}}_p \boldsymbol{X}_p^\flat$, where $\boldsymbol{X}_p^\flat$ denotes the embedded input and $\boldsymbol{\theta}_p$ is the learnable parameters with respect to $\boldsymbol{X}_p^\flat$. The $\boldsymbol{\mathcal{W}}_p$ and $\boldsymbol{\mathcal{V}}_p$ are the decomposition and reconstruction operators that transform the input graph signal $\boldsymbol{X}_p^\flat$ from and to the vertex domain.

Different from a set of independent static graphs, dynamic graphs are captured on an evolving timeline. Therefore, we preserve the intra-connections of the graph sequence in a global learnable variable $\Theta$. At time $t$, we initialize the framelet coefficients $\theta[t]^{(0)}$ with their most recent best estimation before $t$, i.e., $\theta[:t]^{(n)}$, from $\Theta$. For instance, the initial $\theta$ with respect to node $p$ reads $\theta_p[t]^{(0)} = \Theta_p$. Figure 1 demonstrates the update procedure of the global framelet transform with a sample global graph of 5 nodes. Suppose the subgraph at batch $t$ contains the first 3 of the 5 nodes. A UFGCONV trains $\theta[t]$ to represent these three nodes. The parameter values are initialized with the best estimated of $\{\Theta_1, \Theta_2, \Theta_3\}$ recorded before $t$. The optimized model is deployed for further prediction tasks. Meanwhile, $\Theta$ updates the first three parameters by $\{\theta_1[t]^{(n)}, \theta_2[t]^{(n)}, \theta_3[t]^{(n)}\}$.

The workflow of SPEDGNN is summarized in Figure 1 and Algorithm 1. A *temporal spectral transform* first processes the input raw data to the spectral domain that encodes long-range time dependency. With the adaptive power method SVD, a group of stable principal patterns can be extracted, which is functioned similarly to a stacked efficient linear self-attention mechanism. Next, a *message-memory* module enhances the short-term interactions of events and generates a comprehensive node representation of batched subgraphs. The topology of subgraphs records the interactions among node entities for a global graph framelet network to learn. Based on this, we estimate the main algorithm has a linear computational complexity $\mathcal{O}(Nd\log(d))$ to the number of events $N$, which proves that SPEDGNN is efficient with a small time and space complexity (See Appendix C).

## 4 Numerical Examples

We carry out experiments on three bipartite graph datasets (**Wikipedia**, **Reddit**, and **MOOC**)[13, 32] for link prediction and node classification tasks [33]. Both transductive and inductive settings are examined in link predictions. We leave implementation details in Appendix E.

A fair comparison is made with JODIE [13], DYREP [34], and TGN [4]. Classic methods (e.g., TGAT and DEEPWALK) that significantly underperform the baseline methods are excluded. For model training and evaluation, we assume the interactions of a graph are given until the last timestamp in batch $t$ and make predictions on the timestamps of batch $t+1$ and $t+2$, where predictions on the former set provide the validation scores, and the latter guides the test scores. Note that our training follows PyTorch Geometric [35] and makes a more strict data acquirement criterion than TGN, where

---

**Algorithm 1:** SPEDGNN: Spectral Dynamic Graph Neural Network

---

**Input**  : raw sequential data $X$
**Output** : label prediction $Y$

1  Initialization: adaptive coefficient $C$; global $\Theta$
2  **Adaptive power method SVD** $\widetilde{X} \leftarrow X(X^\top X)^q \tilde{R}^{-1} C$;
3  **for** *batch* $p \leftarrow 1$ **to** $M - 2$ **do**
4      $h_p[t] \leftarrow \textbf{msg}(e_i[t]) \| \textbf{mem}(e_i[: t])$;
5      $\mathcal{G}_p \leftarrow (A_p, X_p \leftarrow \textbf{FC}(h_p[t]))$;
6      $H_p \leftarrow \textbf{UFGCONV}(A_p, X_p, \theta_p)$;
7      $Y_p \leftarrow \text{Predictor}(H_p)$;
8      $\Theta_p \leftarrow \theta_p$;
9      $Y_{p,\text{val}}, Y_{p,\text{test}} \leftarrow \text{Predictor}(H_{p+1}), \text{Predictor}(H_{p+2})$;
10     **Update**: score($Y_{\text{val}}$), score($Y_{\text{test}}$).
11 **end**

---

**Table 1:** Performance of link prediction over 10 repetitions.

| | Model | # parameters | Wikipedia | | Reddit | | MOOC | |
|---|---|---|---|---|---|---|---|---|
| | | | precision | ROC-AUC | precision | ROC-AUC | precision | ROC-AUC |
| transductive | DYREP | $920 \times 10^3$ | $94.67_{\pm 0.25}$ | $94.26_{\pm 0.24}$ | $96.51_{\pm 0.59}$ | $96.64_{\pm 0.48}$ | $79.84_{\pm 0.38}$ | $81.92_{\pm 0.21}$ |
| | JODIE-RNN | $209 \times 10^3$ | $93.94_{\pm 2.50}$ | $94.44_{\pm 1.42}$ | $97.12_{\pm 0.57}$ | $97.59_{\pm 0.27}$ | $76.68_{\pm 0.02}$ | $81.40_{\pm 0.02}$ |
| | JODIE-GRU | $324 \times 10^3$ | $96.38_{\pm 0.50}$ | $96.75_{\pm 0.19}$ | $96.84_{\pm 0.39}$ | $97.33_{\pm 0.25}$ | $80.29_{\pm 0.09}$ | $84.88_{\pm 0.30}$ |
| | TGN-GRU | $1,217 \times 10^3$ | $96.73_{\pm 0.09}$ | $96.45_{\pm 0.11}$ | $98.63_{\pm 0.06}$ | $98.61_{\pm 0.03}$ | $83.18_{\pm 0.10}$ | $83.20_{\pm 0.35}$ |
| | SPEDGNN-MLP (ours) | $170 \times 10^3$ | $97.02_{\pm 0.06}$ | $96.51_{\pm 0.08}$ | $98.19_{\pm 0.05}$ | $98.15_{\pm 0.06}$ | $82.40_{\pm 0.24}$ | $85.55_{\pm 0.17}$ |
| | SPEDGNN-GRU (ours) | $376 \times 10^3$ | $97.44_{\pm 0.05}$ | $97.15_{\pm 0.06}$ | $98.69_{\pm 0.09}$ | $98.66_{\pm 0.12}$ | $84.50_{\pm 0.10}$ | $86.88_{\pm 0.09}$ |
| inductive | DYREP | $920 \times 10^3$ | $92.09_{\pm 0.28}$ | $91.22_{\pm 0.26}$ | $96.07_{\pm 0.34}$ | $96.03_{\pm 0.28}$ | $79.64_{\pm 0.12}$ | $82.34_{\pm 0.32}$ |
| | JODIE-RNN | $209 \times 10^3$ | $92.92_{\pm 1.07}$ | $92.56_{\pm 0.87}$ | $93.94_{\pm 1.53}$ | $95.08_{\pm 0.70}$ | $77.17_{\pm 0.02}$ | $81.77_{\pm 0.01}$ |
| | JODIE-GRU | $324 \times 10^3$ | $94.93_{\pm 0.15}$ | $95.08_{\pm 0.70}$ | $92.90_{\pm 0.03}$ | $95.14_{\pm 0.07}$ | $77.82_{\pm 0.17}$ | $82.90_{\pm 0.60}$ |
| | TGN-GRU | $1,217 \times 10^3$ | $94.37_{\pm 0.23}$ | $93.83_{\pm 0.27}$ | $97.38_{\pm 0.07}$ | $97.33_{\pm 0.11}$ | $81.75_{\pm 0.24}$ | $82.83_{\pm 0.18}$ |
| | SPEDGNN-MLP (ours) | $170 \times 10^3$ | $94.27_{\pm 0.05}$ | $93.28_{\pm 0.05}$ | $97.49_{\pm 0.01}$ | $97.34_{\pm 0.02}$ | $82.54_{\pm 0.08}$ | $85.23_{\pm 0.09}$ |
| | SPEDGNN-GRU (ours) | $376 \times 10^3$ | $96.60_{\pm 0.01}$ | $95.70_{\pm 0.02}$ | $97.47_{\pm 0.05}$ | $97.10_{\pm 0.09}$ | $82.35_{\pm 0.06}$ | $83.67_{\pm 0.06}$ |

† The top three are highlighted by **First**, **Second**, **Third**.

the latter has access to all previous data when loading node neighbors, including those in the same test batch. Such an operation reveals the true connections to predict, and the test scores of TGN reported by Rossi et al. [4] are higher than in this paper.

**Prediction Performance.** Table 1 reports the performance of link prediction tasks. SPEDGNN constantly outperforms JODIE and DYREP with RNN, and achieves at least comparable performance to JODIE and TGN-GRU with a small volatility. It is noteworthy that JODIE-GRU outperforms the original JODIE-RNN by Kumar et al. [13]. The performance gain of GRU over RNN explains to some extent the rare outperformance of TGN over SPEDGNN-MLP, not to mention that MLP is simpler than any recurrent unit. This statement is confirmed by the performance of SPEDGNN-GRU. When the GRU module is employed in the memory layer, the highest performance score is almost always observed over different datasets, learning tasks, and evaluation metrics.

In addition to the transductive and inductive link prediction tasks, we also conduct node classification. The model performance is evaluated by the average ROC-AUC scores, which better fit the extremely imbalanced nature of node classes. The results reported in Table 2 confirm that SPEDGNN outperforms all baselines, especially with the GRU module.

**Table 2:** ROC-AUC of node classification

| Model | Wikipedia | Reddit | MOOC |
|---|---|---|---|
| DYREP | $84.59_{\pm 2.21}$ | $62.91_{\pm 2.40}$ | $69.86_{\pm 0.02}$ |
| JODIE-RNN | $85.38_{\pm 0.08}$ | $61.68_{\pm 0.01}$ | $66.82_{\pm 0.05}$ |
| JODIE-GRU | $87.90_{\pm 0.09}$ | $64.30_{\pm 0.21}$ | $70.23_{\pm 0.09}$ |
| TGN-GRU | $88.95_{\pm 0.07}$ | $61.49_{\pm 0.01}$ | $70.32_{\pm 0.13}$ |
| SPEDGNN-MLP (ours) | $88.37_{\pm 0.03}$ | $64.94_{\pm 0.07}$ | $69.52_{\pm 0.08}$ |
| SPEDGNN-GRU (ours) | $90.32_{\pm 0.05}$ | $65.28_{\pm 0.05}$ | $71.08_{\pm 0.02}$ |

**Table 3:** Training speed for link prediction

| Model | Wikipedia | Reddit | MOOC |
|---|---|---|---|
| DYREP | $20.1s_{\pm 0.6s}$ | $139.3s_{\pm 0.1s}$ | $78.34s_{\pm 0.6s}$ |
| JODIE-RNN | $17.4s_{\pm 2.0s}$ | $121.8s_{\pm 0.3s}$ | $62.64s_{\pm 0.1s}$ |
| JODIE-GRU | $16.9s_{\pm 1.1s}$ | $131.6s_{\pm 1.5s}$ | $58.82s_{\pm 2.2s}$ |
| TGN-GRU | $24.9s_{\pm 0.3s}$ | $128.1s_{\pm 2.2s}$ | $78.11s_{\pm 0.7s}$ |
| SPEDGNN-MLP (ours) | $9.87s_{\pm 0.1s}$ | $63.3s_{\pm 1.1s}$ | $38.41s_{\pm 0.5s}$ |
| SPEDGNN-GRU (ours) | $12.5s_{\pm 0.3s}$ | $83.6s_{\pm 0.1s}$ | $49.20s_{\pm 0.1s}$ |

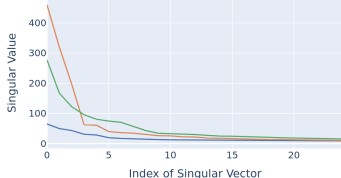

**Figure 3:** The distribution of the largest 25 singular values of the hidden representation by SPEDGNN, TGN, and JODIE.

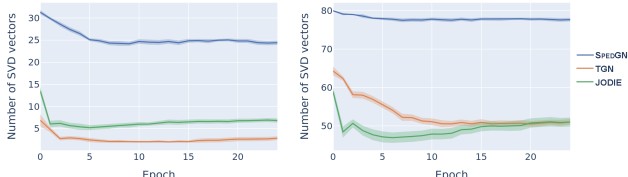

**Figure 4:** The number of singular vectors that provides 50% (left) or 90% (right) of total variance by SPEDGNN, TGN, and JODIE. SPEDGNN constantly includes more vectors to achieve the same level of total variance.

**Computational Efficiency.** Table 3 evaluates model efficiency by the training speed per epoch. Compared to the baseline models, the training speed per epoch of SPEDGNN is shorter with $50\%$ ahead on the largest dataset **Reddit**. It confirms SPEDGNN's long-sequence computational privilege analysed in Section 2. In contrast, the comparable performance by TGN-GRU is achieved at the cost of doubling the training time to fit the model with seven times more learnable parameters. On the other hand, both variants of JODIE require a significantly longer training time than SPEDGNN, not to mention that their performance cannot constantly stay at the top tier.

**Well-conditioned Spectral Node Embedding.** We validate the energy balancing effect of the proposed SPEDGNN by investigating the distribution of hidden embedding's eigenvalues of different models. Recall that in Section 2 we demonstrated with a 2-dimensional toy example that the normalized spectral transform projects input features to well-conditioned representations, which requires a smoother decision boundary that is easier to fit by a classifier. For a higher dimensional feature representation, we describe the smoothness of the decision boundary by the decay of the associated condition number $\lambda_i/\lambda_{\min}$, or the eigenvalues $\lambda_i$.

We made the comparison on the optimized hidden representation of the test samples in **Wikipedia**. The distribution and total variance of condition number are visualized in Figure 3 and Figure 4, respectively. According to Figure 3, JODIE and TGN concentrate the most variance in the first few directions. Eigenvalues of the associated hidden representations decrease drastically after the first 3 or 4 epochs. Such a fast reduction of condition numbers indicates that the analyzed hidden features are highly-correlated, which gives rise to the concentration of the variance of the feature space on the first few principal components. As it challenges the classifier to find the optimal model to fit a rough decision boundary, such a circumstance with concentrated feature energy is not favored. In contrast, SPEDGNN finds a more separable hidden representation of test samples with slowly decayed singular values. As shown in Figure 4, the embedding by SPEDGNN constantly disperses the total variation in a larger number of vectors, while JODIE and TGN pick a few features to undertake most variations after the first few epochs.

## 5 Related work

### 5.1 Efficient Self-Attention

The transformer is well-known for its powerful learning ability [16, 36–39]. However, the self-attention mechanism at the core of a transformer framework requires quadratic time and memory complexity, which hinders the model's scalability. A handful of recent works discuss potential improvements in the efficiency of model memory or computational cost when the input dimension is of a fixed size that is considerably large.

The prominent efficient transformer methods fall into three directions. First, prior knowledge compress or distill the self-attention architecture to a sparse attention matrix by pre-defining strides convolutions [40, 41] or assuming patchwise patterns [19, 42]. Some recent study also considers replacing fixed patterns with a learnable scheme that efficiently identifies chunks or clusters [43–45]. The data-driven learning procedure introduces extra flexibility to the division of the patches, blocks, or receptive fields, but the core idea of attention localization remains.

The second approach simultaneously accesses multiple tokens through a global memory module. The target is to distill the input sequence with a limited number of inducing points (or memory) [46, 47]. Compared to the first approach of patching input tokens, inducing points break down the strict concept of token entities and make parameterizations on the global memory of token mixers.

The third emerging technique avoids explicitly computing the full contextual matrix of the self-attention mechanism through kernelization [28, 48] or low-rank approximation [21–23]. The projection is usually conducted on the lengthy sequence dimension that ignores the chronicle order of sequence when computing attention scores. However, the global view in compress helps the attention mechanism to manage the overall picture of the sequence on top of token-wise correlations. As is investigated by a recent study [49], the substitution of matrix decomposition to the self-attention mechanism is critical for learning the global context.

## 5.2 Graph Structure Embedding

GNNs have seen a surge in interest and popularity for dealing with irregular graph-structured data that traditional deep learning methods such as CNNs fail to manage. Common to most GNNs and their variants is the graph embedding through the aggregation of neighbor nodes in a way of message passing [50–52]. As a key ingredient for topology embedding, graph convolutions correspond to spatial methods and spectral methods that operate on node space [53, 54] or on a pseudo-coordinate system that is mapped from nodes through some transform (typically Fourier) [55].

Due to the intuitive characteristics of spatial-based methods which can directly generalize the CNNs to graph data with convolution on neighbors, most GNNs fall into the category of spatial methods [54, 56–63]. Many other spatial methods broadly follow the message passing scheme with different neighborhood aggregation strategies, but they inherently lack expressivity [61, 64, 65].

In contrast, spectral-based graph convolutions [25, 55, 66–73] convert the raw signal or features in the vertex domain into the frequency domain. Spectral-based methods have already been proven to have a solid mathematical foundation in graph signal processing [74], and the vastly equipped multi-scale or multi-resolution views push them to a more scalable solution of graph embedding. Versatile Fourier [66, 67, 75], wavelet [69] and framelets [25] transforms have also shown their capabilities in graph representation learning. Of these transforms, Fourier transforms is particularly one of the most popular ones and the work in [76] gave a detailed review of how the Fourier transform enhances neural networks. In addition, with fast transforms being available in computing strategy, a big concern related to efficiency is well resolved.

## 5.3 Temporal Encoding of Dynamic Graphs

Recurrent neural networks (RNNs) are considered exceptionally successful for sequential data modeling, such as text, video, and speech [77–79]. In particular, Long Short Term Memory (LSTM) [80] and Gated Recurrent Unit (GRU) [15] gains great popularity in application. Compared to the Vanilla RNN, they leverage a gate system to extract memory information, so that memorizing long-range dependency of sequential data becomes possible.Later, the Transformer network [16] designs an encoder-decoder architecture with the self-attention mechanism, so as to allow parallel processing on sequential tokens. The self-attention mechanisms have achieved state-of-the-art performance across all NLP tasks [16, 33] and even some image tasks [20, 81].

For dynamic GNNs, it is critical to consolidate the features along the temporal dimension. Dynamic graphs consist of discrete and continuous two types according to whether they have the exact temporal information [82]. Recent advances and success in static graphs encourage researchers and enable further exploration in the direction of dynamic graphs. Nevertheless, it is still not recently until several approaches [34, 83–85] were proposed due to the challenges of modeling the temporal dynamics. In general, a dynamic graph neural network could be thought of as a combination of static GNNs and time series models which typically come in the form of an RNN [86–88]. The first DGNN was introduced by Seo et al. [86] as a discrete DGNN and Know-Evolve [89] was the first continuous model. JODIE [13] employed a coupled RNN model to learn the embeddings of the user/item. The work in [90] learns the node representations through two joint self-attention along both graph neighborhood and temporal dynamics dimensions. The work in [91] was the first to use RNN to regulate the GCN model, which means to adapt the GCN model along the temporal dimension at every time step rather than feeding the node embeddings learned from GCNs into an RNN. TGAT [92] is notable as the first to consider time-feature interactions. Then Rossi et al. [4] presented a more

generic framework for any dynamic graphs represented as a sequence of time events with a memory module added in comparison to [92] to enable short-term memory enhancement.

## 6   Discussion

This work analyzes the versatile spectral transform in capturing the evolution of long-range time series as well as graph topology. We investigate a particular dynamic system of continuous-time dynamic graphs (CTDGs) to find its robust representation. In particular, we implement iterative SVD approximations to encode the long-range feature evolution of the dynamic graph events, which acts a similar role as multiple layers of a low-rank self-attention mechanism. The proposed transform has linear complexity of $\mathcal{O}(Nd\log(d))$ for a CTDG with $N$ events of $d$ dimensions. The short-term memory in learning is enhanced for dynamic events by a learnable scheme, such as MLP or GRU. A multi-level and multi-scale fast transform of global spectral graph convolution is then employed for topological embedding, which allows sufficient scalability and transferability in learning dynamic graph representation. The final event embeddings are well-conditioned and the algorithm requests fewer calculation resources. The proposed SPEDGNN shows competitive performance on real dynamic graph prediction tasks.

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

# A   Message-Memory Aggregator

A continuous-time dynamic graph records the temporal evolution by a sequence of events. After the spectral transformer in Section 3.2, the event-based instances are projected to a spectral domain of the feature dimension, where the long-term dependencies are well-encoded. However, the intrinsic structural information is waiting for embedding.

To this end, a memory 'window' gets involved that divides multiple batches of subgraphs. This operation allows zooming into a small range of events and generating graph snapshots for further topological embedding. Since dynamic graphs are a generalized version of non-Euclidean time series, the learning procedure requires an enhancement of short-term memory. That is, the current state of an underlying node is believed to be closely connected to recent messages of the same entity (node) from previous states. Thus, instead of treating graph intervals as discrete slices, a message-memory aggregator practices node embedding with short-range information inheritance [4, 13].

Given an event $e_i[t]$ at time $t$ with respect to node $n_i$, we name it a *message* of $n_i$ at time $t$, denoted as $\mathbf{msg}(e_i[t])$. In addition, if the node was previously recorded active, we use $\mathbf{mem}(e_i[:t])$ to represent the past information, or *memory*, of $n_i$ prior to time $t$. The memory module $\mathbf{mem}(\cdot)$ refreshes constantly with the latest messages to capture the dynamic nature of graph interactions. When a new event $e_i[t]$ is recorded, the updated memory at time $t$ is

$$\mathbf{mem}(e_i[t]) = f(\mathbf{mem}(e_i[:t]), \mathbf{msg}(e_i[t]))$$

with a trainable function $f(\cdot)$. Depending on when the node $i$ was previously recorded, the last memory can be found before $t-1$. Also, it is possible to recall memories from more than one step away.

We thus describe $n_i[t]$'s state by its hidden memory $h_i[t]$ at time $t$, which concatenates $\mathbf{msg}(e_i[t])$ and $\mathbf{mem}(e_i[:t])$, i.e.,

$$h_i(t) = \mathbf{concat}(\mathbf{msg}(e_i[t])\|\mathbf{mem}(e_i[:t]))). \tag{5}$$

The embedding for an interactive event $e_{ij}(t)$ between two nodes $n_i$ and $n_j$ is similar, which reads

$$h_i(t) = \mathbf{concat}(\mathbf{msg}(e_{ij}[t])\|\mathbf{mem}(e_i[:t])\|\mathbf{mem}(e_j[:t])). \tag{6}$$

# B   Graph Framelet Transforms

This section briefs the graph framelet transforms [25, 93], which fast approximation of framelet coefficients is the foundation for an efficient global UFGCONV algorithm in Section 3.4.

**Framelet System.**   A framelet is defined by two key elements: a *filter bank* $\boldsymbol{\eta} := \{a; b^{(1)}, \ldots, b^{(K)}\}$ and a set of *scaling functions* $\Psi = \{\alpha; \beta^{(1)}, \ldots, \beta^{(K)}\}$. We name $a$ the low-pass filter and $b^{(k)}$ the $k$th high-pass filter with $k = 1, \ldots, K$. The two sets of filters respectively extract the approximated and detailed information of the input graph signal in the framelet domain. The choice of filter masks results in different tight framelet systems. This work considers the *Haar-type* filter with one high pass, i.e., $K = 1$. For $x \in \mathbb{R}$, it defines

$$\widehat{\alpha}(x) = \cos(x/2) \ \text{ and } \ \widehat{\beta^{(1)}}(x) = \sin(x/2).$$

For other choices, such as *linear* and *quadratic* filters, we refer the interested readers to [93].

**Framelet Transform.**   Framelet transform divides an input signal to multiple channels by a set of low-pass and high-passes *framelet bases*. For a specific nodes $p$, its undecimated framelet bases at *scale level* $l = 1, \ldots, J$ reads

$$\begin{aligned}
\boldsymbol{\varphi}_{l,p}(v) &:= \textstyle\sum_{\ell=1}^{n} \widehat{\alpha}\left(\tfrac{\lambda_\ell}{2^l}\right) \overline{\boldsymbol{u}_\ell(p)}\boldsymbol{u}_\ell(v), \\
\boldsymbol{\psi}_{l,p}^{(k)}(v) &:= \textstyle\sum_{\ell=1}^{n} \widehat{\beta^{(k)}}\left(\tfrac{\lambda_\ell}{2^l}\right) \overline{\boldsymbol{u}_\ell(p)}\boldsymbol{u}_\ell(v), \quad k = 1, \ldots, K.
\end{aligned} \tag{7}$$

Here the eigenpairs $\{(\lambda_\ell, \boldsymbol{u}_\ell)\}_{\ell=1}^{n}$ of the graph Laplacian $\mathcal{L}$ plays a key role in embedding graph topology. The $\boldsymbol{\varphi}_{l,p}(v), \boldsymbol{\psi}_{l,p}^{(k)}(v)$ with $v \in \mathcal{V}$ are named the low-pass and the $k$th high-pass framelet basis, respectively. They project input signals to a transformed domain as *framelet coefficients*. Given a signal $\boldsymbol{x}$, $\langle \boldsymbol{\varphi}_{l,p}, \boldsymbol{x} \rangle$ and $\langle \boldsymbol{\psi}_{l,p}^{k}, \boldsymbol{x} \rangle$ are the corresponding low-pass and high-pass framelet coefficients for node $p$ at scale $l$. They respectively record the approximated global information and detailed local information of the graph signal.

**Fast Graph Framelet Transform.** Define $\mathcal{W}_{k,l}$ the *framelet decomposition operator* as a set of orthonormal framelet bases at $(k,l) \in \{(0,J)\} \cup \{(1,1),\dots,(1,J),\dots(K,1),\dots,(K,J)\}$. Essentially, calculating $\mathcal{W}$ at the low-pass and the $k,l$th high-pass requires

$$\mathcal{W}_{0,J} = \boldsymbol{U}\widehat{\alpha}\left(\frac{\Lambda}{2}\right)\boldsymbol{U}^\top \text{ and } \mathcal{W}_{k,l} = \boldsymbol{U}\widehat{\beta^{(k)}}\left(\frac{\Lambda}{2^{l+1}}\right)\boldsymbol{U}^\top \quad \forall l = 0,\dots,J. \tag{8}$$

To avoid time-consuming eigendecomposition to the graph Laplacian, we consider $m$-order Chebyshev polynomials for a fast approximation of the filter spectral functions. Denote the $m$-order approximation of $\alpha$ and $\{\beta^{(k)}\}_{k=1}^K$ by $\mathcal{T}_0$ and $\{\mathcal{T}_k\}_{k=1}^K$, respectively. The framelet decomposition operator $\mathcal{W}_{r,j}$ is approximated by

$$\mathcal{W}_{k,l} = \begin{cases} \mathcal{T}_0\left(2^{-R}\mathcal{L}\right), & l = 1, \\ \mathcal{T}_r\left(2^{R+l-1}\mathcal{L}\right)\mathcal{T}_0\left(2^{R+l-2}\mathcal{L}\right)\dots\mathcal{T}_0\left(2^{-R}\mathcal{L}\right), & l = 2,\dots,J. \end{cases} \tag{9}$$

The dilation scale $H$ satisfies $\lambda_{\max} \leq 2^H\pi$.

## C Complexity Analysis

In this section, we show the spectral transformer is efficient with a small time and space complexity. In particular, we analyze the computational complexity of SPEDGNN following Algorithm 1 by estimating the cost for the three main computational units: power method SVD for the entire training data, and MLP and UFGCONV (graph framelet convolution) for graph batches.

**Time Complexity.** For a dynamic graph with $N$ events (edges) and $d$ edge features, the computational cost for power method SVD is $\mathcal{O}(Nd\log(d))$ [94]. The MLP has cost $\mathcal{O}(N)$ in total. For all $M$ batches, the framelet convolution (UFGConv) has the complexity of $\mathcal{O}(\sum_{p=1}^M n_p S_p \log_2(\lambda_p/\pi)F)$ where $n_p, S_p$ are the number of edges and sparsity of the $p$th batched graph, $\lambda_p$ is the largest eigenvalue of the corresponding graph Laplacian [25], and $F$ is the number of the node features. In practice, the $n_p$ for each batched graph can be set as $n/M$, and we suppose $S_p$ and $\lambda_p$ are bounded by constants. The total computational cost of SPEDGNN is $\mathcal{O}\left(N(d\log(d)+F)\right)$.

**Space complexity.** For the power method SVD, the memory cost is $\mathcal{O}(Nd)$. The MLP with $L+1$ fully connected layers needs memory $\mathcal{O}(\sqrt{N/M} \times h_1 + h_1 \times h_2 + \dots + h_l \times h_{L+1})$, where $h_i$ denotes the hidden unit of the $i$th layer, and $h_{L+1}$ is determined by the output dimension. Suppose each layer has the same number of hidden neurons $h$, then MLP has the space complexity $\mathcal{O}(h\sqrt{N/m} + h^2 L)$. The memory cost of framelet convolution is $\mathcal{O}(NF)$. Then, the total space complexity of SPEDGNN is $\mathcal{O}\left(N(d+F) + h\sqrt{N/M} + h^2 L\right)$.

**Parameter number.** The trainable network parameters appear mainly in MLP and UFG-CONV. Similar to the space complexity analysis, SPEDGNN has a total number of $\mathcal{O}\left(h\sqrt{N/M} + h^2 L + NF\right)$ parameters.

The empirical computational efficiency of SPEDGNN is evaluated by comparing against competitor models with link prediction tasks in Table 3 in the main paper.

## D Dataset Descriptions

The experiments are conducted on three bipartite graph datasets: **Wikipedia**, **Reddit** and **MOOC** [13, 32].

- **Wikipedia** has users and Wikipedia pages as the two sets of nodes. An edge is recorded when a user edits a page. The dataset selects the $1,000$ most edited pages and frequent editing users who made at least 5 edits. The dataset contains $9,227$ nodes and $157,474$ edges in total, and each event is described by 172 features.

- **Reddit** divides two sets of nodes as users and subreddits (communities). An interaction occurs when a user posts a message to a subreddit. The datasets samples the $1,000$ most active

subreddits as nodes along with the $10,000$ most active users. In total, the dataset contains $11,000$ nodes and $672,447$ edges. All events are recorded as 172 edge features by the LIWC categories [95] by the text of each post.

- **MOOC** records students and courses of the "Massive Open Online Course" learning platform. An interaction occurs when a student enrolls in the course. The dataset consists of $7,047$ students, 97 courses and $411,749$ interactions. Specifically, $4,066$ state changes are recorded implying action that a student drops out of a course.

## E    Implementation Details

The best-reported performance of all the methods is tuned with PyTorch on NVIDIA® Tesla V100 GPU with 5,120 CUDA cores and 16GB HBM2 mounted on an HPC cluster.

### E.1    Training Setup

We follow the pseudo-code of Algorithm 1 and design SPEDGNN accordingly. In the spectral attention module, we approximate truncated SVD with some largest modes with $q$-iteration. The specific number of nodes is selected as the smallest number between 50 and 100 such that the spectral norm error is less than $0.1$. The memory batches are processed by fully connected layers, and the prepared subgraphs are then processed by UFGCONV with Haar-type filters at dilation factor $2^l$ to allow efficient transforms. To train a generalized model that is robust to small disturbance, in the validation set we randomly add 50% negative samples at each epoch. The same negative sampling procedure is conducted in the test set, except that all the samples are deterministic. We follow the design convention of negative sampling to promise non-trivial training and prediction. The hyper-parameters of baseline models, unless specified, are fixed to the best choice provided by their authors. For all the models, we fix the batch size at $1,000$ with a maximum of 200 epochs for both datasets. Any employed neural network overlays either 2 or 3 layers, and the memory dimension, node embedding dimension, and time embedding dimension are selected from $\{100, 150, 200\}$ respectively. To make the comparison as fair as possible, the number of parameters of each model corresponding to the fine-tuned hyperparameters are reflected in Table 1. The optimal learning rate is tuned from the range of $\{1e-4, 5e-5\}$, and the weight decay is fixed at $1e-2$. The training process is optimized by ADAMW [96]. All the datasets follow the standard split and processing rules as in [4, 13]. The average test accuracy and its standard deviation come from 10 runs of random initialization.

### E.2    Model Availability

We use publicly available programs to implement baseline methods, which are available at:

- JODIE [13]: `https://github.com/srijankr/jodie`;
- DYREP [34]: implemented by `https://github.com/twitter-research/tgn`;
- TGN [4]: `https://github.com/twitter-research/tgn`.

The PyTorch version implementation for SPEDGNN is at `https://github.com/bzho3923/GNN_SpedGNN`.

## F    Ablation Study

In addition to the comparison against baseline methods, we also designed an ablation study to justify the choice of the temporal spectral transform and the sequential network. For the former, we set up two different data encoders in terms of using the adaptive SVD (ASVD). spectral transform or not (RAW). The output data matrix is then used for graph slicing and then batch training. For the sequential network in message-memory aggregation, we compare MLP and GRU modules, which are also the two choices we evaluated in the last two experiments. Lastly, we compare the choice of using UFG or not.

The models are validated on Wikipedia for transductive link prediction, following the same setups aligned with Table 1. The hyper-parameters are fixed to the optimal results from the best performed SPEDGNN in the earlier baseline comparison experiment.

**Table 4:** Average performance of ABLATION study on Wikipedia with link prediction.

| Module | precision | ROC-AUC |
|---|---|---|
| RAW+MLP+UFG | 96.71±0.10 | 96.20±0.20 |
| RAW+GRU+UFG | **97.18±0.03** | **96.84±0.04** |
| ASVD+MLP+UFG | **97.02±0.06** | **96.51±0.08** |
| ASVD+GRU+UFG | **97.44±0.05** | **97.15±0.06** |
| ASVD+GRU | 96.54±0.12 | 96.12±0.09 |

Table 4 reports the average precision and ROC-AUC with the first, second, and third highest scores highlighted in red, violet, and black respectively. Under both evaluation metrics, including UFG module always result in a noticeably better performance. For the choice of GRU or MLP for message-memory aggregation, it confirms the finding in Section 4 that GRU improves prediction performance over simpler MLP modules, but the outperformance is not as high when MLP collaborates with the adaptive SVD module, which achieves comparable scores to the RAW+GRU setting. Overall, the choice of combining ASVD+GRU+UFG in our model reaches the highest prediction scores. These observations justify that well-conditioned data due to spectral transform benefits various models for prediction and classification tasks.

