# OpenReview forum: "Well-conditioned Spectral Transforms for Dynamic Graph Representation"
_logconference.io/LOG/2022/Conference — LoG 2022 Poster_

### Official Review · Reviewer_6nDG · 2022-10-10

**Overall Score:** 8
**Confidence:** 4

**Review:**

Description:
The paper proposes a spectral method, SPEDGNN, for continuous time dynamic graphs. Firstly the method selects the useful components of features from the input embedding space using a learnable approximation of the power SVD method. The graphs are batched by time and a “memory module” is used to learn features from the current and previous timesteps. These features at each timestep are then given to the spectral(framelet graph convolution) module. This module updates its parameters from the optimized parameters in the set of the previous timesteps. The authors show that the current method gives better results than the baselines. Also the power SVD based method is able to keep the information spread over the spectrum.

Strengths:
1) The paper has a valid motivation of keeping the decision boundaries in latent space linear and conserving useful(but possibly rare) information
2) The reported results are better than baselines indicating the model’s efficacy

Weaknesses/Improvements:
1) Since the method uses multiple components an ablation study should be conducted analyzing the effect of each. These could be for eg: 1) SPEDGNN without the power SVD method 2) SPDGNN without memory module 3) SPDGNN without UFGConv
2) Regarding the approximation of the inverse of the upper triangular matrix(\tilde{R}^{-1}) in the QR decomposition using a learnable upper triangular parameter W: How does the method ensure W=\tilde{R}^{-1}? Is there some regularization term to minimize for eg: \norm{XW - Q}?


Comments/Limitation/Concerns:
1) In line 108 the authors mention the method “prevents important rare patterns from being removed due to their small energy”. Could the authors provide some case studies from the datasets showing the same?
2) The testing method is different from the baseline(TGN). It seems the authors have trained/implemented baselines for evaluating. It is possible that the baselines were not optimized enough and the results are a bit off. Hence could the authors provide results for the proposed method on the same settings that baselines use?


Typos:
1) In the appendix the authors mention “SPEGER” as the methods name. Shouldn’t this be “SPEDGNN”?

---

### Official Review · Reviewer_ksQt · 2022-10-22

**Overall Score:** 6
**Confidence:** 4

**Review:**

This paper uses adaptive power method based method + framelet graph convolution model. The model achieves competitive performance dynamic system of continuous-time dynamic graphs tasks.

Pros:
1. The spectral method proposed in this paper have a good condition number.
2. The  proposed transform has linear complexity.
3.  Achieve competitive performance on the  link prediction tasks.

Cons:
1. From equation 2, the author uses a learnable matrix to calculate the $\tilde{X}$, however, in Algorithm 1, $\tilde{X}$ is calculated by $R^{-1}$, do the author use the learnable matrix for approximate $\tilde{X}$? if yes, where is the learnable matrix?
2. I find the Algorithm 1 part is hard to follow, e.g. the definition of $\textbf{msg}$ and $\textbf{mem}$ are in the appendix and may confuse the reader. I suggest the author make the Spectral Transform part short and add more details about the algorithm in the main page.
3. The authors report the theoretical complexity, it would be great to report the realistic running time of different GNN methods.

---

### Official Review · Reviewer_HahT · 2022-10-22

**Overall Score:** 6
**Confidence:** 3

**Review:**

**Description**

This paper proposes SpeedGNN for continuous-time dynamic graph problems. It uses SVD to capture long-range evolution for dynamic graphs. Then uses a message-memory module for short-term interaction. In the end, use global framelet graph convolution for topological embedding. The authors experimentally demonstrate the effectiveness and efficiency of the proposed model on benchmark datasets.

**Strengths**
-  The paper is well written. It outlines clear motivation and provides detailed descriptions.
- The paper performs better than baselines on link prediction and node classification indicating generalisation across tasks.

**Weakness**
- The paper consists of various moving parts. It would be better if the author show the effectiveness of each module in their approach.

 **Comments**
- Adaptive Data Augmentation on Temporal Graphs (NeurIPS 21): The results of this paper was not included even though it performs better. Why is this the case?

---

### Meta-Review · Area_Chair_wqyZ · 2022-11-11

**Confidence:** 5
**Recommendation:** Accept

**Meta Review:**

The paper proposes a new method for continuous-time dynamic graph problems. Authors designed an architecture using a learnable approximation of the power SVD to model the long-range evolution of dynamic graphs. A message-memory is then used to generate node representations of batched subgraphs. Finally, the topology of these subgraphs is used to learn a global graph framelet net. The method has attractive linear complexity.

Reviewers broadly agree on the novelty of the approach, the clarity of the paper and of the proposed experiments as well as on the performances for link prediction tasks. Reviewers pointed to potential weaknesses of the used sub-modules that compose the architecture (power SVD, framelet network) to which the author convincingly answered by providing ablation studies in the rebuttal round.

Summary and decision: Quality of presentation, competitiveness of performances on link prediction tasks and ablation studies outweigh minor concerns, I therefore recommend to accept this paper.

---

### Decision · Program_Chairs · 2022-11-23

Accept (Poster)